# An IoT-Enabled E-Nose for Remote Detection and Monitoring of Airborne Pollution Hazards Using LoRa Network Protocol

**DOI:** 10.3390/s23104885

**Published:** 2023-05-19

**Authors:** Kanak Kumar, Shiv Nath Chaudhri, Navin Singh Rajput, Alexey V. Shvetsov, Radhya Sahal, Saeed Hamood Alsamhi

**Affiliations:** 1Department of Electronics Engineering, Indian Institute of Technology (BHU), Varanasi 221005, India; kanakkumar.rs.ece18@iitbhu.ac.in (K.K.); shivnathchaudhri.rs.ece17@iitbhu.ac.in (S.N.C.); 2Department of Electronics and Communication Engineering, Santhiram Engineering College, Nandyal 518501, India; 3Department of Smart Technologies, Moscow Polytechnic University, 107023 Moscow, Russia; a.shvetsov@vvsu.ru; 4Department of Transport, North-Eastern Federal University, 677000 Yakutsk, Russia; 5School of Computer Science and IT, University College Cork, T12 K8AF Cork, Ireland; rsahal@ucc.ie; 6Faculty of Computer Science and Engineering, Hodeidah University, Al Hodeidah P.O. Box 3114, Yemen; 7Faculty of Engineering, Ibb University, Ibb P.O. Box 70270, Yemen; s.alsamhi.rs.ece@iitbhu.ac.in

**Keywords:** Internet of Things (IoT), long range (LoRa), airborne pollution hazard, intelligent gas sensor system (IGSS), low-power wide-area network (LPWAN)

## Abstract

Detection and monitoring of airborne hazards using e-noses has been lifesaving and prevented accidents in real-world scenarios. E-noses generate unique signature patterns for various volatile organic compounds (VOCs) and, by leveraging artificial intelligence, detect the presence of various VOCs, gases, and smokes onsite. Widespread monitoring of airborne hazards across many remote locations is possible by creating a network of gas sensors using Internet connectivity, which consumes significant power. Long-range (LoRa)-based wireless networks do not require Internet connectivity while operating independently. Therefore, we propose a networked intelligent gas sensor system (N-IGSS) which uses a LoRa low-power wide-area networking protocol for real-time airborne pollution hazard detection and monitoring. We developed a gas sensor node by using an array of seven cross-selective tin-oxide-based metal-oxide semiconductor (MOX) gas sensor elements interfaced with a low-power microcontroller and a LoRa module. Experimentally, we exposed the sensor node to six classes i.e., five VOCs plus ambient air and as released by burning samples of tobacco, paints, carpets, alcohol, and incense sticks. Using the proposed two-stage analysis space transformation approach, the captured dataset was first preprocessed using the standardized linear discriminant analysis (SLDA) method. Four different classifiers, namely AdaBoost, XGBoost, Random Forest (RF), and Multi-Layer Perceptron (MLP), were then trained and tested in the SLDA transformation space. The proposed N-IGSS achieved “all correct” identification of 30 unknown test samples with a low mean squared error (MSE) of 1.42 × 10^−4^ over a distance of 590 m.

## 1. Introduction

The World Health Organization (WHO) reports that indoor air pollution (IAP) harms 3.8 million people every year [1]. It is caused by occupant activities such as cooking, smoking, using electronic equipment, or using consumer items, or by emissions from construction materials inside houses or buildings. Carbon monoxide (CO), volatile organic compounds (VOCs), particulate matter (PM), aerosols, and other biological pollutants are some of the harmful pollutants that are found in the indoor ambient air of buildings [2]. In addition, it has been reported that poor indoor air quality (IAQ) may harm people’s health by increasing disorders that are associated with airborne pollution present in indoor ambient air [3,4]. Thus, airborne pollution detection and monitoring of the presence of various hazardous VOCs, gases, and smokes are essential to mitigate poor air quality in indoor spaces.

Low-power wide-area network (LPWAN) technologies have attracted considerable interest and played a significant role in popularizing Internet of Things (IoT)-based applications. Depending on the deployment area, most LPWANs operate in the unlicensed industrial, scientific, and medical (ISM) bands at 169, 433, 868/915 MHz, and 2.4 GHz [5]. A LoRa-based network link is a wireless communication technology that uses long-range, low-power radio frequency signals to transmit data over a long distance. These LPWAN technologies include NB-IoT, Sigfox, and LoRa. It has been demonstrated that LoRa offers advantages in terms of ultralong transmission, high stability, ultralow power, and low cost [6]. LoRa-based network links are designed to provide reliable and cost-effective communication for IoT devices in various applications, including smart cities, industrial automation, agriculture, and environmental monitoring [5].

LoRa-based network links have two main components, i.e., LoRa nodes and LoRa gateways. The LoRa nodes are small, low-power devices that can be embedded in IoT sensors and devices to transmit data wirelessly. The LoRa gateways act as bridges between the LoRa nodes and the Internet, receiving data from the nodes and forwarding them to a cloud-based server or an IoT platform. The characteristics of several network technologies used for the IoT are shown in Table 1. This study introduces the LoRa WAN protocol-based pollution hazard detection and monitoring system, comprising a microcontroller and up to eight sensors and operating remotely. Further, advanced two-stage analysis space transformation-based approaches were also used to enhance the performance and efficiency of the proposed networked intelligent gas sensor system (N-IGSS) [7,8,9].

The literature review showed a growing interest in environmental monitoring using IoT and cloud-based technologies. Numerous environmental factors, including temperature, humidity, light, noise level, CO, and NO_2_, were monitored indoors and outdoors on a university campus using an environmental monitoring system [12]. This system included the Raspberry Pi and MICS-4514 sensors. An IoT-based indoor environmental quality (IEQ) assessment system employing an Arduino Uno module and economical sensors like the DHT22 was developed in an experimental investigation [13]. An IoT-based measurement system with simple pressure, temperature, humidity, and gas sensors (BME680, DHT22, and MQ5) interfaced with an Arduino microcontroller has also been reported [14]. In a similar experiment, Kureshi et al. used temperature, humidity, a VOC sensor (BME680), and PM 2.5 and PM 10 sensors (SDS011) for air quality index (AQI) calculation. They used this information to increase awareness about indoor air pollution’s hazards to human health [4]. Numerous gases, including CO_2_, NO_2_, ethanol, methane, and propane, were identified using the gas sensors MICS-6814 in an IoT-based system [15] developed for real-time IAQ monitoring.

An investigation of the emissions generated by different incense sticks and sparklers, including particles and benzene, was reported by Werner et al. [16]. Utilized indoors, they had a significantly unfavorable impact on the air quality. The indoor AQI was further explored by using multiple low-cost PM sensors for effective compliance with the U.S. Environmental Protection Agency (EPA) [17]. The variation in the concentration ranges of CO, CO_2_, total volatile organic compounds (TVOCs), and PM 2.5 over a period of 24 h was monitored in indoor environments by Capua et al., and a decision-making algorithm was used to manage the IAQ [18]. Rebecca et al. described the particles and gases released by burning Arabian incense and investigated the reactions of in vitro human lung cells to incense smoke [19]. A study on mosquito-repellent incense was conducted by Wang et al. The incense was found to generate excessive gaseous and particle pollutants, as permitted under GB3095-2012 rules and the WHO advice [20]. Furthermore, green IoT for eco-friendly applications was discussed by the authors of [21] and opportunistic routing (OR) was widely adopted in wireless sensor networks (WSNs) running asynchronous duty-cycled MAC protocols by the authors of [22]. 

In the recent literature, researchers have only considered AQI and used specific sensors to detect and monitor a limited number of pollutant gases and PM concentrations for the evaluation of indoor air quality. The attention towards various VOCs, gases, and smokes in indoor environments is lacking. Furthermore, an electronic nose (e-nose) is a device that mimics the human olfactory system to identify and analyze various VOCs, gases, and smokes present in the environment. E-noses have been widely used for detecting and monitoring air pollution hazards, including industrial emissions, traffic emissions, and indoor air pollutants. The proposed approach for pollution hazard detection and monitoring using e-noses is to provide real-time monitoring of airborne pollutants at a centrally located RDPS. The data collected at the RDPS are analyzed by leveraging artificial intelligence-based algorithms and models to identify the classes of pollutants of interest [7,8,9]. Pollution hazard detection networks are designed to detect, monitor, and predict the class and level of environmental pollution. Such networks can be categorized based on the sensors, spatial coverage, and data transmission methods. In addition, people around the world utilize the Internet and GSM mobile-based networks, which require high power and paid subscriptions [6].

In this paper, we propose a networked intelligent gas sensor system (N-IGSS) using a LoRa network-link-based IoT platform which can be operated in private places such as industrial settings, mines, etc., and it can operate well even when there is no Internet or GSM mobile-based network connectivity. To the authors’ knowledge, this is the first work to have integrated a LoRa network link to develop a N-IGSS. The proposed N-IGSS can be used in many practical LoRa-based applications, such as hospital management, air quality monitoring, agriculture, smart cities, and industrial management. Therefore, the proposed N-IGSS will produce economic, social, and environmental impacts.

### 1.1. Motivation and Contributions

We were motivated by the goal of a low-power, low-cost solution for wide monitoring of airborne hazards across remote locations. The proposed solution addresses the challenge of monitoring airborne hazards using Internet connectivity, which consumes significant power. Furthermore, the proposed system can operate independently by utilizing LoRa-based wireless networks that do not require Internet connectivity, thus reducing power consumption and providing extended battery life.

The proposed N-IGSS comprises an array of low-cost, MOX-based gas sensor elements integrated as the gas sensor node. In this experiment, we considered six types of VOCs, gases, and smokes released by burning tobacco, paint, carpet, incense sticks, and alcohol. Our sensor node was placed indoors to capture and transmit sensor responses via the LoRa network link to the remotely placed LoRa gateway (remote data-processing station) for further analysis and predictions for real-time operations. We processed the sensor response data at the RDPS. The block schematic diagram of our proposed N-IGSS is shown in Figure 1. In the proposed N-IGSS, an array of seven-element tin-oxide-based gas sensor elements, along with one digital temperature and humidity sensor, is used for the detection of six classes of VOCs, gases, and smokes. The respective signature patterns were analyzed using our proposed two-stage analysis space transformation method, which closely mimics the human olfactory system [23]. The utilities of this work are highlighted as follows:

An N-IGSS is proposed to detect and monitor airborne pollution hazards in indoor ambient air using an RDPS.For the first time, a LoRa WAN networking link protocol was used for real-time networked operation of e-noses.The proposed N-IGSS was designed using a two-stage analysis space transformation method to ensure that the classifier models delivered high performance.

### 1.2. Paper Structure

This paper is structured as follows: An introduction to the N-IGSS and some prominent LoRa network application scenarios are presented in Section 1. In Section 2, the architecture of our proposed N-IGSS using LoRa network hardware, subsequent experiments, and contextual details of the tools and processes are presented. The results and discussions are placed in Section 3. In Section 4, conclusions and future scope are presented.

## 2. Materials and Methods

In this section, we present the method used to develop the proposed N-IGSS, which is integrated with the LoRa networking link protocol for hazardous pollution detection in real time at an RDPS using artificial-intelligence-based algorithms and models.

### 2.1. The Contextual Background of the N-IGSS

The proposed system comprises three parts, as shown in Figure 2a–c.

Physical gas sensor node (e-nose): The sensor node is the onsite part of the proposed N-IGSS, which consists of an array of seven tin oxide (MOX)-based gas sensor elements, named MQ 2, MQ 3, MQ 5, MQ 6, MQ 7, MQ 8, and MQ 135, and one temperature and humidity sensor, DHT 22, interfaced with a microcontroller and a LoRa module, as shown in Figure 2a. These MOX sensors are cross-sensitive and respond to multiple VOCs, gases, and smokes with different sensitivities and generate unique signature patterns for various VOCs, gases, and smokes in different types of pollutants. 

The LoRa link: The LoRa link is a wireless communication technology designed to enable long-range, low-power communication between devices. The LoRa link operates in the unlicensed radio spectrum, which means that it can be used without needing a license or paying subscription fees. The main function of the LoRa link is to provide a reliable and efficient communication link for IoT devices. It is particularly well-suited for applications that require low data rates and long-range communication, as shown in Figure 2b.

RDPS: We received the real-time transmissions of the sensor node responses at the other end of the LoRa link using the LoRa gateway. LoRa is a transceiver module, and we were able to use the same module as a transmitter and receiver according to the configuration code. At this RDPS, we processed the gas sensor array responses using our proposed two-stage space transformation method to achieve high performance. The captured gas sensor array responses were first preprocessed via this method to transform the raw sensor responses into a suitable analysis space using the standardized linear discriminant analysis (SLDA) method. In the second stage, different types of classifiers, namely AdaBoost, XGBoost, RF, and MLP, were designed in the SLDA analysis space to efficiently classify the samples of the six classes of VOCs, gases, and smokes, as shown in Figure 2c.

### 2.2. The Gas Sensor Node Prototype

The gas sensor node consists of seven tin-oxide-based MOX-based gas sensor elements and one temperature and humidity sensor, which generate real-time signature patterns of the considered VOCs, gases, and smokes released when burning the considered types of pollutant materials. The sensor node was placed indoors to transmit sensor responses to the analyzed VOCs, gases, and smokes using the LoRa network link protocol. These wirelessly transmitted sensor array responses are received at an RDPS (LoRa link gateway). The electrical characteristics of the components used for the sensor node prototype development are shown in Table 2.

The connection diagrams of the sensors and the LoRa transmitter and receiver with the microcontroller are shown in Figure 3a,b.

We interfaced the sensors and LoRa modules with their respective microcontrollers to create the sensor node and LoRa receiver gateway of the proposed N-IGSS. It communicates with the sensors and transmits the data through a serial peripheral interface (SPI) to the LoRa transceiver. The LoRa module SX1278 is a 137 MHz to 525 MHz long-range low-power transceiver module which can be used as both a transmitter and a receiver. The sensor node is powered using a 5 V (DC) supply. The LoRa module (SX1278) is supplied with 3.3 V by voltage-dividing the microcontroller’s 5 V supply. The basic circuit diagram of the N-IGSS is depicted in Figure 4a, showing the sensor node, and Figure 4b, showing the receiver gateway at the RDPS.

LoRa consists of an antenna that transmits the sensor values through the microcontroller to the receiver node via radio waves. In our case, we used a 915 MHz antenna in the ISM band designated for Asia. A physical view of the fabricated N-IGSS for pollution hazard detection and monitoring is shown in Figure 5a,b.

The LoRa transceiver at the receiver gateway is coded exclusively to function as the gateway. This gateway can be connected to the Internet to upload real-time sensor responses to the cloud. Multiple nodes can also be used to create a more effective and larger network of IGSS.

### 2.3. Experimental Setup

In ambient conditions, we conducted our experiments in a closed, semi-furnished room of 114 square meters (1226.64 square feet). We deployed the N-IGSS sensor node at a fixed height of 2 feet and captured real-time responses of the sensor array for the analyzed smokes of tobacco, paints, carpet, alcohol, and incense sticks. In our experiments, we employed the SX1278 LoRa module, which can only operate on the 868 MHz (European ISM) and 915 MHz frequencies (American ISM). To adhere to Federal Communications Commission (FCC) rules, we selected the 915 MHz band. 

Initially, the gas sensor system was purged with ambient air for 10 min to obtain the baseline signature pattern of the sensor array responses under steady-state conditions. Once the sensor array responses became steady, the array was exposed to one of the considered classes of VOCs, gases, or smokes for 20 min at a rate of 15 samples per minute. After 20 min of exposure, the gas sensor array was again purged with ambient air until the sensor responses returned to the baseline signature patterns. This process was repeated for all the considered classes of VOCs, gases, and smokes, and the raw sensor array response dataset was captured in its totality. 

In this experiment, the sensor node was first heated up for 15 min while keeping the gas chamber closed, and the steady state of the sensor responses was determined. Consequently, the gas chamber inlet was opened for the next 20 min, and samples of the considered analyte were captured at the RDPS. Then the analyte exposure was stopped, and for the next 30 min, the gas chamber was purged with fresh ambient air to ensure that the sensor responses returned to the baseline conditions. The same procedure was repeated to capture samples of the other the analytes.

Accordingly, each experimental phase continued for 65 min, raw sensor responses were captured, and the same process was repeated in all experiments. Therefore, the experiment took 390 min (65 min × 6 types). All sensor responses returned to baseline responses throughout the experiment, and no sensor poisoning occurred. During this period, 1800 samples were captured at a sampling rate of 15 samples per minute. Further details of the dataset and the samples collected are given in Table 3.

The samples belonged to the six classes of VOCs, gases, and smoke released by burning different pollutants in indoor ambient conditions. The dataset consisted of 300 samples of each class: ambient air, tobacco, paints, carpet, incense sticks, and alcohol. The captured dataset was then segregated into training and testing datasets. The training dataset consisted of 295 × 6 = 1770 samples, and 5 × 6 = 30 samples were used for testing purposes. Finally, the testing dataset was separated from the data used for training or validation of the classifiers at any stage.

A prototype LoRa network was implemented and deployed in a typical urban environment for airborne pollution hazard detection and monitoring and to evaluate the efficacy of the proposed LoRa network. In the experiment, the LoRa node was gradually moved away from the LoRa gateway’s location while sending uplink packets, and the received signal strength was measured continuously. A LoRa node transmits a data packet every 4 s, and the packet is then received at the LoRa gateway. Since the LoRa gateway can communicate across channels, we received the sensor responses correctly at the remote data-processing station.

### 2.4. Contextual Background of Analysis Space Transformation

The proposed N-IGSS is based on performance enhancement approaches in which suitable classifiers are designed in the analysis space transformation domain. It has been reported that a classifier performs better when it is trained in a suitable transformation space where the dataset shows well-separated clusters with good intercluster separation [8]. In the following subsections, we describe the processes in greater detail.

#### 2.4.1. Data Preprocessing

This work used the standardized linear discriminant analysis (SLDA) space transformation method. The SLDA transforms raw data into respective space transformation domains [7,24,25,26,27,28]. The first three principal components contained 98.88% of the information, using which we obtained a 3D scatter plot which showed well-separated clusters with good intercluster separation [8], as shown in Figure 6.

Standardized linear discriminant analysis (SLDA) is a statistical method that can be used for dimensionality reduction. It is a linear transformation process applied to the input data, which transforms the raw sensor responses into new components to maximize the separation between different classes.

The process of SLDA space transformation is described in the following steps:Calculate the mean vector, *m_i_* (*i* = 1,2,3…) of each class of a dataset Scatter matrix within the class
(1)$$S_{W}=\sum_{i=1}^{c}S_{i}$$
where S_W_= data points within each class that deviate from their respective class.
(2)$$S_{i}=\sum_{x\in D_{i}}^{n}\left(x-m_{i}\right){\left(x-m_{i}\right)}^{T}$$
where *S_i_* = scatter matrix of each class, *x* = data point, *m_i_* = mean vector, and *T* = transpose matrix.

(3)$$m_{i}=\frac{1}{n_{i}}\sum_{x\in x_{i}}^{n}x_{i}$$

Calculate the covariance matrix by adding the scaling factor (1/(*N* − 1)) to the within-class scatter matrix
(4)$$\sum_{i}=\frac{1}{N_{i}-1}\sum_{x\in D_{i}}^{n}\left(x-m_{i}\right){\left(x-m_{i}\right)}^{T}$$
(5)$$\mathrm{And}\;S_{W}=\sum_{i=1}^{c}(N_{i}-1)\sum_{i}$$
where *N_i_* = sample size of the VOC class (here = 300 × 6). We can now drop (*N_i_* − 1) because all classes have an equal sample size.

Scatter matrix between each class (*S_B_*):
(6)$$S_{B}=\sum_{i=1}^{c}N_{i}\left(m_{i}-m\right){\left(m_{i}-m\right)}^{T}$$
where *m* = overall mean, *m_i_* = sample mean, and *N_i_* = sample size of the respective class.

Compute the eigenvectors and eigenvalues:
(7)$${A=S}_{W}^{-1}S_{B}$$
(8)AV=λV
where λ = eigenvalue and *V* = eigenvector of same eigenvalue

Project the data onto the new subspace:
Y = X × W(9)
where X = *n*-dimension matrix representation of the n samples and Y = transmitted *n* × k dimensional samples in the new subspace.

We preprocessed the raw sensor response dataset in this experiment using the standardized linear discriminant analysis (SLDA) space transformation method. The SLDA method transforms raw data into its respective space transformation domains [7,20,23,24]. The first three principal components contained 98.88% of information, using which we obtained a 3D scatter plot which showed well-separated clusters with good intercluster separation, as shown in Figure 7. It can be observed that the clusters belonging to the six classes of VOCs/gases/odors were shown clearly in the SLDA-transformed dataset.

#### 2.4.2. Design of the Classifiers

The raw sensor responses were first transformed into the SLDA domain during preprocessing. SLDA is an effective method for feature enhancement. We used all seven principal components (PCs) to train and test the classifier used in the IGSS without any information loss. In the SLDA transformation space, better-performing classifiers were then designed. The transformed dataset consisted of 1800 samples, with each vector consisting of seven elements. The transformed dataset was then segregated into two parts, i.e., the training and testing datasets, consisting of 1770 and 30 samples in the SLDA-transformed domain, respectively. Furthermore, we used four popular classifiers for the experiment, namely AdaBoost, XGBoost, RF, and MLP. AdaBoost and XGBoost are good for boosting, RF for handling large datasets, and MLP for handling complex nonlinear relationships. The classifier process is shown in Figure 8.

The AdaBoost algorithm is a boosting algorithm that combines multiple weak classifiers to form a robust classifier. The design of an AdaBoost classifier involves selecting several hyperparameters, such as the number of estimators, the learning rate, and the random state, to optimize the model’s performance. The number of estimators is the number of weak classifiers that combine to form a robust classifier. The learning rate is a hyperparameter that controls the contribution of each weak classifier to the final model. The random state is a seed value that ensures the reproducibility of the results. In this case, we set the random state to 1. The AdaBoost classifier is shown in Figure 9.

The design of an XGBoost classifier involves selecting several hyperparameters, such as the learning rate, number of estimators, maximum depth of the tree, minimum child weight, gamma, regularization alpha, number of threads, and cross-validation. The learning rate is a hyperparameter that controls the contribution of each tree to the final model. The number of estimators is the number of decision trees created in the ensemble. The XGBoost classification model is shown in Figure 10.

Random Forest builds many decision trees and combines their outputs to make predictions. The design of a Random Forest classifier involves selecting several hyperparameters, such as the number of estimators, criterion, learning rate, random state, and cross-validation. The number of estimators is the number of decision trees created in the forest. The criterion is a hyperparameter that determines the function used to measure the quality of a split. The random state is a hyperparameter that controls the randomness of the algorithm. The Random Forest classification model is shown in Figure 11.

This MLP classifier has a relatively small number of hidden layers and a moderate batch size, which makes it computationally efficient and suitable for medium-sized datasets. Using the ReLU activation function and adaptive learning rates can help to improve the model’s performance, while using cross-validation can help to reduce overfitting and improve generalization. The ReLU activation function is defined as f(x) = max (0, x), which means that the output of the activation function is the maximum between the input value and zero. The model adapts the learning rate of each weight during training based on the historical gradient information, which can lead to faster convergence and better performance than fixed learning rates. An epoch is defined as one pass through the entire training dataset. The number of iterations required for the model to converge may be less than 100. The proposed MLP classification model is shown in Figure 12.

The performance parameters taken while designing the classifiers AdaBoost, XGBoost, RF, and MLP are presented in Table 4.

## 3. Results and Discussion

AdaBoost, XGBoost, RF, and MLP classifiers were designed and tested in this experiment. The MLP trained using the SLDA-transformed data achieved an “all correct” classification accuracy across the 30 test samples taken from the six classes of VOCs, gases, and smokes. The proposed IGSS is intended to be portable, easy to use, and affordable in real-world scenarios.

### 3.1. The LoRa Network Link Performance

In this experiment, we tested the LoRa channel’s performance in an indoor ambient setting with several wall and roof obstructions. The signal strength of the LoRa networking link was measured as the received signal strength indicator (RSSI) in decibels (dB). It was observed that the RSSI was about −60 to −80 dB when the transmitted signal became erratic. However, the received signal exhibited errors as the RSSI decreased with increasing distance. The signal was completely lost when the RSSI was −164 dB. Accordingly, the maximum experimental range achieved for the LoRa network link for indoor operation of the N-IGSS was 590 m.

### 3.2. Performance of the Proposed N-IGSS for Airborne Pollution Hazard Detection

In this experiment, we trained and tested four different classifiers. Among these, the AdaBoost, XGBoost, and Random Forest (RF) classifiers achieved 96.67%, 96.67%, and 96.67% classification accuracy, respectively, for the 30 unknown test samples. On the other hand, the MLP classifier achieved 100% classification accuracy for all 30 test samples. While classifying the considered VOCs, gases, and smokes, we compared the prediction error for each test sample. The error ranged between 1.42 × 10^−4^ and 1.24 × 10^−2^, with an average mean-squared error (MSE) of 2.66 × 10^−3^ for 30 unknown test samples. The classification performance of the best-trained classifier is shown in Table 5 and Figure 13. Furthermore, the classification performance of the MLP classifier, trained and tested in the SLDA domain using 30 unknown test samples of the considered VOCs, gases, and smokes, is shown in Figure 14.

For further clarity, the confusion matrix of the classification performance of the MLP classifier is also shown in Figure 15.

The classification accuracy was calculated using popular error matrices, i.e., mean squared error (MSE) and mean absolute error (MAE). The MLP classifier was the best-performing classifier. The differences between the actual and predicted values for the six classes of VOCs, gases, and smokes were evaluated using MSE and MAE as the performance parameters, as shown in Table 6. For the sake of further clarity, the confusion matrix of the classification performance of the MLP classifier is shown in Figure 14, which shows the “all correct” classification of the 30 unknown samples taken from the testing dataset and not used for training of the classifier models in the SLDA transformation domain. 

## 4. Conclusions and Future Work

In this work, we proposed an N-IGSS for airborne pollution hazard monitoring. It consists of seven tin-oxide-based gas sensor nodes interfaced with a microcontroller and a LoRa network link for airborne pollution hazard detection and monitoring in real time. The developed N-IGSS link has a useful link length of 590 m for real-time sensor response reception at the RDPS. The LoRa module consumes minimal power and provides extended battery life. The system’s ability to detect and monitor airborne pollution hazards with 100% accuracy was demonstrated using 30 test samples. The LoRa and sensor array network allows inexpensive N-IGSS development with limited battery power. The N-IGSS is intended to be portable, easy to use, and affordable in real-world scenarios. The limitations of the proposed N-IGSS are the battery life (a solution could be to connect the N-IGSS to a solar panel) and connectivity range. In future, this work could be scaled up to monitor many other VOCs, gases, smokes, and odors by training the AI model suitably. A wide area network of N-IGSS covering an entire hospital, mall, industrial setting, or an entire smart city could be created, which might consist of thousands to millions of sensor nodes feeding real-time data 24 × 7 to the command-and-control center. The proposed N-IGSS can be scaled up further with many more sensors for AQI monitoring in the short term and for climatic change detection on a long-term basis. 

## Figures and Tables

**Figure 1 sensors-23-04885-f001:**
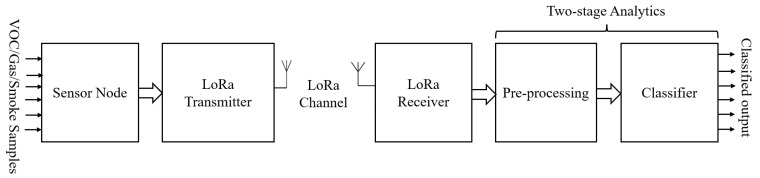
Block schematic diagram of the networked intelligent gas sensor system (N-IGSS).

**Figure 2 sensors-23-04885-f002:**
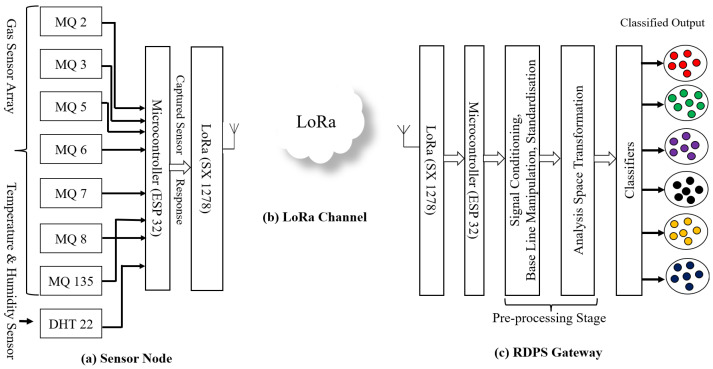
(**a**–**c**) The proposed architecture of the networked intelligent gas sensor system (N-IGSS).

**Figure 3 sensors-23-04885-f003:**
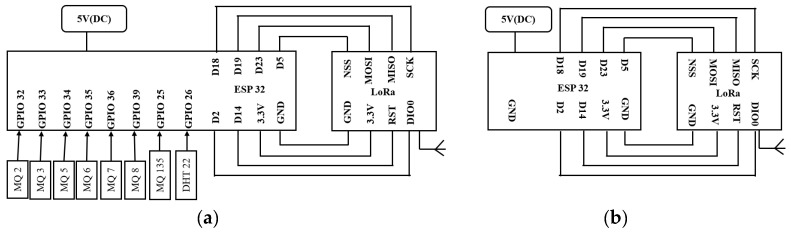
Connection diagram of (**a**) sensor node (transmitter) and (**b**) gateway (receiver module), as interfaced with the microcontroller and LoRa module.

**Figure 4 sensors-23-04885-f004:**
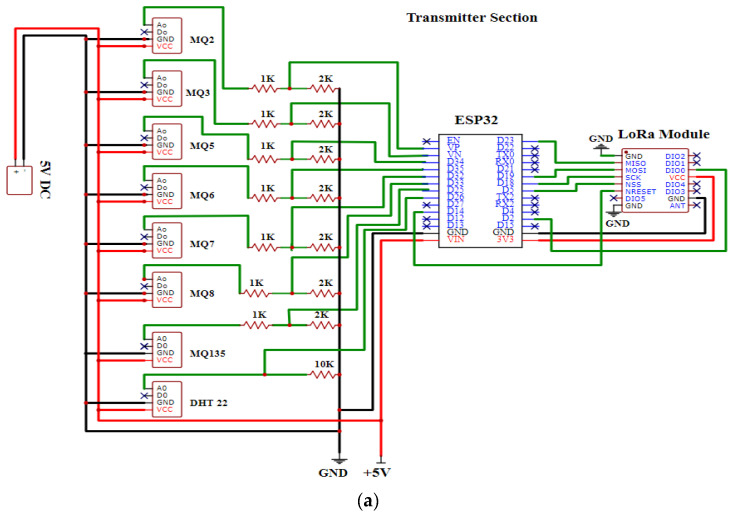
Basic circuit diagram of the N-IGSS: (**a**) sensor node and (**b**) receiver gateway at the RDPS.

**Figure 5 sensors-23-04885-f005:**
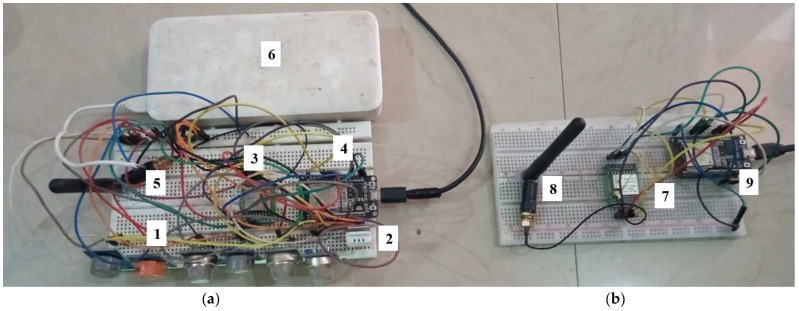
Physical prototype of N-IGSS: (**a**) sensor node, (**b**) receiver gateway at the RDPS. 1: Gas sensor array; 2: temperature and humidity sensor; 3: LoRa module (Tx); 4: microcontroller (Tx); 5: antenna (Tx); 6: power supply (DC, 5 V); 7: LoRa module (Rx); 8: antenna (Rx); 9: microcontroller (Rx).

**Figure 6 sensors-23-04885-f006:**
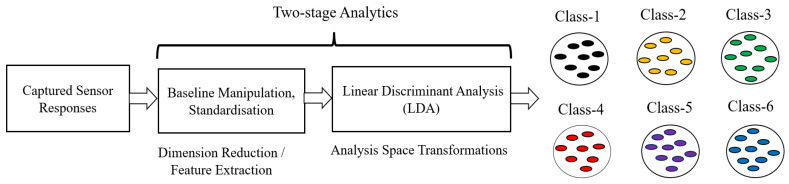
Block schematic diagram of process for obtaining 3D scatter plot.

**Figure 7 sensors-23-04885-f007:**
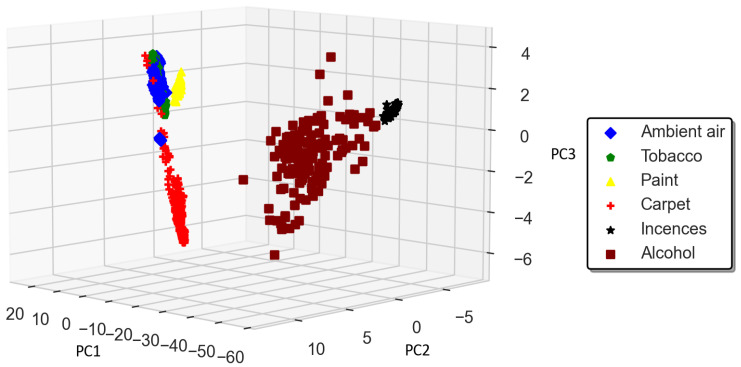
3D scatter plot of SLDA-transformed dataset.

**Figure 8 sensors-23-04885-f008:**
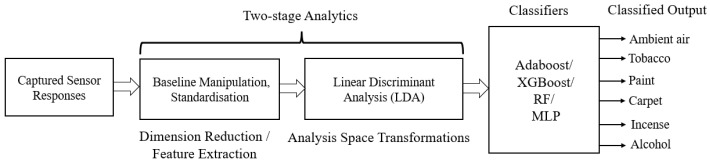
Block schematic diagram of proposed classifiers.

**Figure 9 sensors-23-04885-f009:**
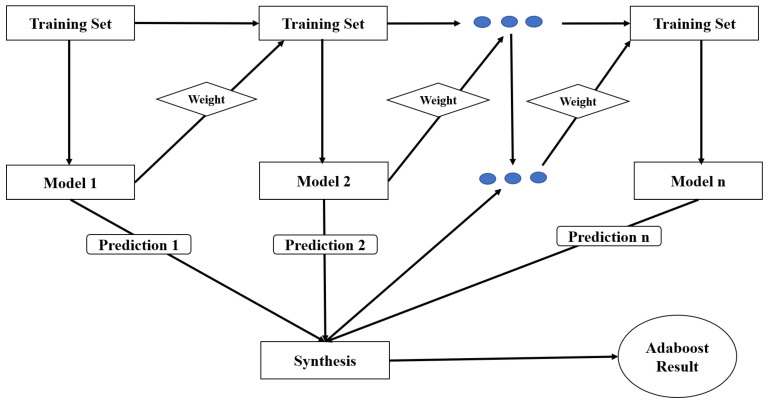
AdaBoost classification model.

**Figure 10 sensors-23-04885-f010:**
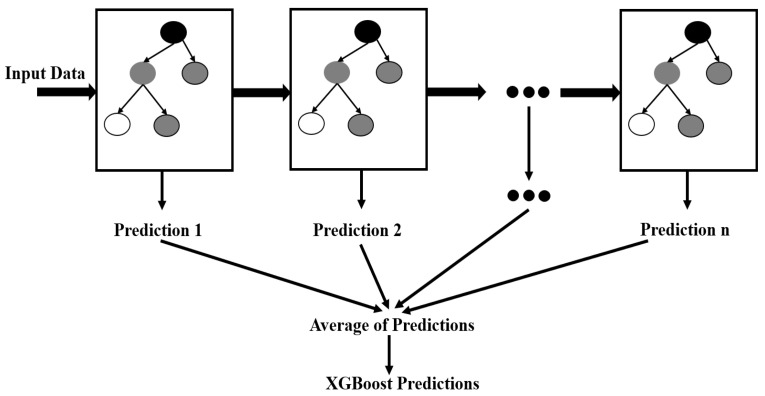
XGBoost classification model.

**Figure 11 sensors-23-04885-f011:**
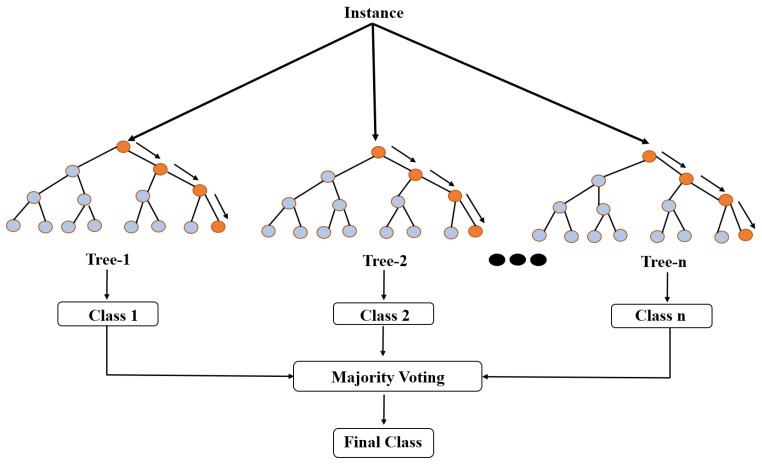
Random Forest classification model.

**Figure 12 sensors-23-04885-f012:**
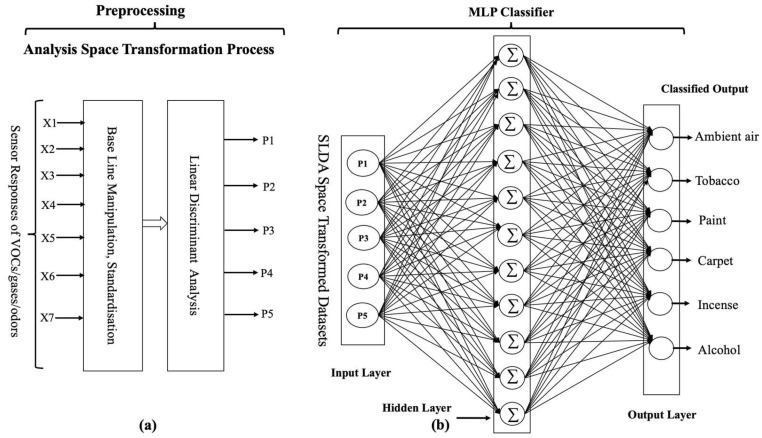
MLP classification model (**a**,**b**).

**Figure 13 sensors-23-04885-f013:**
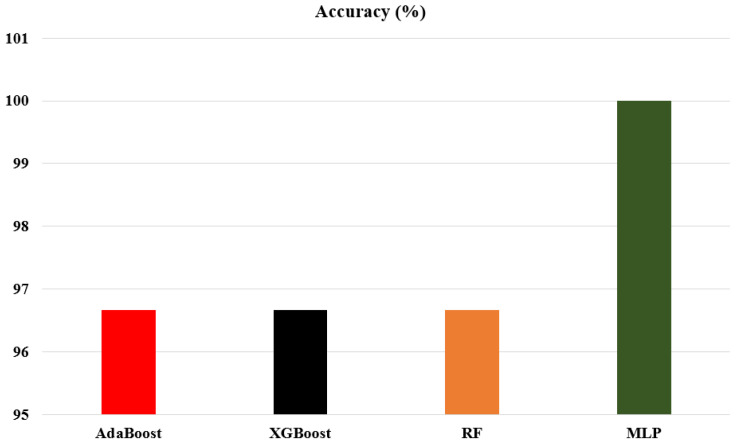
Performance of AdaBoost, XGBoost, RF, and MLP classifiers.

**Figure 14 sensors-23-04885-f014:**
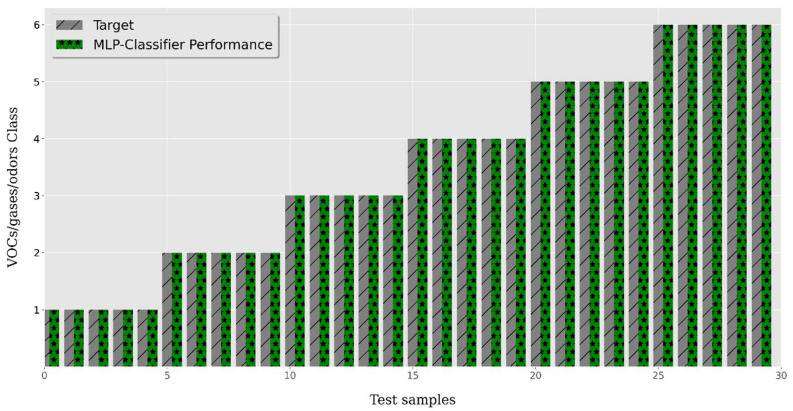
Performance of MLP classifier.

**Figure 15 sensors-23-04885-f015:**
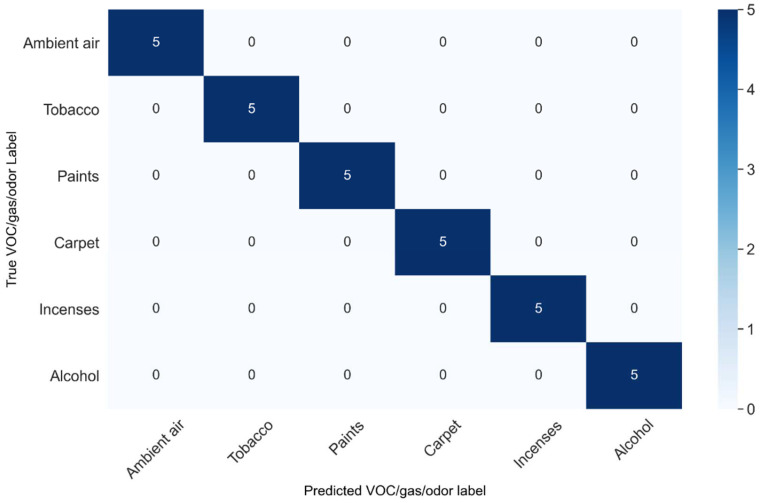
Confusion matrix of the MLP classifier.

**Table 1 sensors-23-04885-t001:** Topology and frequency of the communication technologies [6,10,11].

Network Technologies	Topology	Coverage Range	Power Consumption	Radio Frequency	Tx- Rx Data Size	Limitations/ Advantages
BLE	Ad-hoc	10–100 m	15–30 mA per packet	2.4 GHz–2.4835 GHz	1–3 Mbps	Short range
Wi-Fi	Star	50–100 m	2 to 20 watts	2.4 GHz–5 GHz	1–9608 Mbps	Short distance, high battery power
ZigBee	Mesh	10–100 m	150 mA	868.3 MHz, 902–928 MHz	20–250 kbps	Short distance, maintenance costs too much
Sigfox	Star	20–25 km	78 mA	862–928 MHz	100 bps	High module costs, high battery power
LoRa	Star/ Mesh	10–20 km	32 mA	433 MHz, 860–1020 MHz	290 bps–50 kbps	More extended range, low battery power

**Table 2 sensors-23-04885-t002:** Electrical characteristics of the components used in the device prototype.

List of Components	Input Voltage	Power Ratings
LoRa Module (SX1278)	3.3 V	Tx: 93 mA, Rx: 12.15 mA, standby: 1.6 mA
ESP 32 Microcontroller	5 V	130 mA
ESP 32 GPIO pins	3.3 V	40 mA
MQ Sensor	5 V	150 mA
DHT-22	5 V	2.5 mA

**Table 3 sensors-23-04885-t003:** The distribution of samples.

Raw Materials	Sampling Time (min)	Total Samples	Training Samples	Testing Samples	Class
Ambient air	20	300	295	5	1
Tobacco	20	300	295	5	2
Paints	20	300	295	5	3
Carpet	20	300	295	5	4
Incense	20	300	295	5	5
Alcohol	20	300	295	5	6
Total	120	1800	1770	30	

**Table 4 sensors-23-04885-t004:** Parameter tuning of the AdaBoost, XGBoost, RF, and MLP classifiers.

Classifier	Parameters
AdaBoost	N_estimators:0.5, learning rate: 50, random_state:1, cv = 5
XGBoost	Learning_rate:0.1, n_estimators:1000, max_depth:4, min_child_weight:6, gamma = 0, reg_alpha:0.005, nthread:4, cv = 5
RF	N_estimators:100, criterion: Gini, random_state:1, cv = 5
MLP	Hidden layer sizes = 11, activation function: ReLU, solver: adam, batch size:100, learning rate: adaptive, max iteration: 100, cv = 5

**Table 5 sensors-23-04885-t005:** Performance of different classifiers on SLDA-transformed data.

Classifier	Accuracy (%)
AdaBoost	96.67
XGBoost	96.67
RF	96.67
MLP	100

**Table 6 sensors-23-04885-t006:** MSE and MAE performance of LDA and proposed method.

Class	MSE		MAE	
	LDA	Proposed Method	LDA	Proposed Method
Ambient air	4.62 × 10^−4^	1.42 × 10^−4^	1.003 × 10^−2^	7.53 × 10^−2^
Tobacco	5.38 × 10^−4^	4.51 × 10^−4^	1.74 × 10^−2^	1.41 × 10^−2^
Paints	3.39 × 10^−4^	5.05 × 10^−4^	1.44 × 10^−2^	1.73 × 10^−2^
Carpet	6.53 × 10^−3^	1.53 × 10^−3^	3.02 × 10^−2^	2.29 × 10^−2^
Incense	1.02 × 10^−3^	8.94 × 10^−4^	2.39 × 10^−2^	2.21 × 10^−2^
Alcohol	1.65 × 10^−2^	1.24 × 10^−2^	8.82 × 10^−2^	8.37 × 10^−2^

## Data Availability

The dataset used in this study is available permitted on request.

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
