# Peer review of "An IoT-Enabled E-Nose for Remote Detection and Monitoring of Airborne Pollution Hazards Using LoRa Network Protocol"

_sensors, 2023, doi:10.3390/s23104885_

Round 1
Reviewer 1 Report
The study titled “An IoT Enabled E-nose for Remote Detection and Monitoring of Air Borne Pollution Hazard Using LoRa Network Protocol” proposed a networked intelligent gas sensor system (N-IGSS) using a LoRa network link-based IoT platform which can be operated in private places like in industrial scenarios, mines, etc. Further, the results show that the captured dataset was first pre-processed using the Standardized Linear Discriminant Analysis (SLDA) method. Four different classifiers viz. AdaBoost, XGBoost, Random Forest (RF) and Multi-Layer Perceptron (MLP) were then trained and tested in the SLDA transformation space. Our proposed N-IGSS achieves ‘all correct’ identification of 30 unknown test samples with a low mean squared error (MSE) of 1.42 × 10-4 over a distance of 590 m. There is potential in the research presented in the article, and it should be published after the following revisions have been made.
- Novelty of the research is clear. However, it is needed to correlate research motivation, research problem statement, research aim, research gap, and outcomes in the abstract, introduction, and conclusion sections
- How you see such kind of research after a decade, particularly from the viewpoint of artificial intelligence.
- Limitations and cope of the overall research methodology and opportunities for further research can be highlighted
- It would be helpful to provide a summary of the key advantages and limitations of the LoRa Network Protocol.
- Please provide more details on the experimental setup, such as environmental condition in which experiments were conducted.
- It would be helpful the author providing some more context around the importance and potential impacts of their research approach, by discussing real world use cases or scenarios where this technology could be applied.
- Introduction: it could be further improved by providing more specific details about the proposed pollution hazard detection and monitoring system, including the types of sensors that will be used and how the data will be analyzed.
- Result and discussion: it could benefit from the inclusion of more visual aids such as graphs and tables to support the analysis.
- Result and discussion: It would be helpful to discuss the limitation of experimental setup and provide suggestions for further research in this area.
Author Response
To Reviewer 1
Comments and Suggestions for Authors
The study titled “An IoT Enabled E-nose for Remote Detection and Monitoring of Air Borne Pollution Hazard Using LoRa Network Protocol” proposed a networked intelligent gas sensor system (N-IGSS) using a LoRa network link-based IoT platform which can be operated in private places like in industrial scenarios, mines, etc. Further, the results show that the captured dataset was first pre-processed using the Standardized Linear Discriminant Analysis (SLDA) method. Four different classifiers viz. AdaBoost, XGBoost, Random Forest (RF) and Multi-Layer Perceptron (MLP) were then trained and tested in the SLDA transformation space. Our proposed N-IGSS achieves ‘all correct’ identification of 30 unknown test samples with a low mean squared error (MSE) of 1.42 × 10-4 over a distance of 590 m. There is potential in the research presented in the article, and it should be published after the following revisions have been made.
Author response: Thank you very much for summarising our paper contributions.
Author action: We updated the manuscript by addressing your invaluable comments and suggestion. Authors are highly appreciated your concern and suggestions.
Reviewer 1, concern 1: Novelty of the research is clear. However, it is needed to correlate research motivation, research problem statement, research aim, research gap, and outcomes in the abstract, introduction, and conclusion sections.
Author response: Thank you very much for your concern to improve our manuscript.
Author action: We updated the manuscript by updating abstract, introduction and conclusion as per your invaluable suggestions. We updated the manuscript by highlighting the motivation, problem, and gap in Abstract, introduction (add subsection 1.1 motivation and contributions) and conclusion as shown in the revised version of the manuscript.
Reviewer 1, concern 2: How you see such kind of research after a decade, particularly from the viewpoint of artificial intelligence.
Author response: Thank you very much for your concern.
Author action: We updated the manuscript by highlighting the research after decade in conclusion section in the revised version of the manuscript.
Reviewer 1, concern 3: Limitations and cope of the overall research methodology and opportunities for further research can be highlighted
Author response: Thank you very much for your concern.
Author action: We updated the manuscript by highlighting the limitation and opportunities in conclusion section as shown in the revised version of the manuscript.
Reviewer 1, concern 4: It would be helpful to provide a summary of the key advantages and limitations of the LoRa Network Protocol.
Author response: Thank you very much for your concern.
Author action: we updated the manuscript by highlighting the benefits and limitation of loRa network protocol as shown in motivation and contributions in the revised version of manuscript.
Reviewer 1, concern 5: Please provide more details on the experimental setup, such as environmental condition in which experiments were conducted.
Author response: Thank you very much for your concern.
Author action: We updated the manuscript by adding subsection in 2. Materials and Methods Methodology, i.e., 2.3. Experimental setup as shown in the revised version of manuscript.
Reviewer 1, concern 6: It would be helpful the author providing some more context around the importance and potential impacts of their research approach, by discussing real world use cases or scenarios where this technology could be applied.
Author response: Thank you very much for your concern.
Author action: We update the manuscript by highlighting potention impacts and application as shown in the revised version of the manuscript. The proposed N-IGSS can be used in many practical LoRa-based applications, such as hospital management, air quality monitoring, agriculture, smart cities, and industrial management. Therefore, the proposed N-IGSS will gain an economic, social and environmental impacts.
Reviewer 1, concern 7: Introduction: it could be further improved by providing more specific details about the proposed pollution hazard detection and monitoring system, including the types of sensors that will be used and how the data will be analyzed.
Author response: Thank you very much for your concern.
Author action: We updated the manuscript by providing specific details about proposed pollution hazard detection and monitoring system as shown in the revised version of the manuscript.
An array of seven tin-oxide (MOX)-based gas sensor elements viz. MQ 2, MQ 3, MQ 5, MQ 6, MQ 7, MQ 8, MQ 135 and one temperature and humidity sensor viz. DHT 22 interfaced with a microcontroller and a LoRa module.
The LoRa link is to provide a reliable and efficient communication link for IoT devices. LoRa is a transceiver module, and we can use the same module as a transmitter and receiver according to the configuration code. At this RDPS, we have processed the gas sensor array responses using our proposed two-stage space transformation method to achieve high performance. The captured gas sensor array responses are first pre-processed in this method to transform the raw sensor responses into a suitable analysis space using the Standardized Linear Discriminant Analysis (SLDA) method. In the second stage, different types of classifiers, viz., AdaBoost, XGBoost, RF, and MLP, are designed in the SLDA analysis space to efficiently classify the samples of the six classes of VOCs, gases, and smokes.
Reviewer 1, concern 7: Result and discussion: it could benefit from the inclusion of more visual aids such as graphs and tables to support the analysis.
Author response: Thank you very much for your concern.
Author action: We updated the manuscript by adding figure 13 and more description in table 6. As shown in the revised version of the manuscript.
Reviewer 1, concern 7: Result and discussion: It would be helpful to discuss the limitation of experimental setup and provide suggestions for further research in this area.
Author response: Thank you very much for your concern.
Author action: We updated the manuscript by highlighting the future work and limitation in conclusion section as shown in the revised version of the manuscript.
Reviewer 2 Report
In this manuscript “An IoT Enabled E-nose for Remote Detection and Monitoring of Air Borne Pollution Hazard Using LoRa Network Protocol” is presented.
1. The abstract is quite long. It should be precise. The things which would had been discussed in the introduction section have inserted in the abstract section.
2. Kindly proofread the manuscript for some typo errors.
For example: Word Correction in line 75====efficiency
And so on…
3. Few of the figures are blurry, if authors can improve their quality. For example: Fig. 8.
Proofread is required for typo errors.
Author Response
To Reviewer 2
Comments and Suggestions for Authors
In this manuscript “An IoT Enabled E-nose for Remote Detection and Monitoring of Air Borne Pollution Hazard Using LoRa Network Protocol” is presented.
Author response: Thank you very much for summarising our paper contributions.
Author action: We updated the manuscript by addressing your invaluable comments and suggestion. Authors are highly appreciated your concern and suggestions.
Reviewer 2, concern 1: The abstract is quite long. It should be precise. The things which would had been discussed in the introduction section have inserted in the abstract section.
Author response: Thank you very much for your concern.
Author action: We updated the manuscript by removing unnecessary sentence form abstract to make it more precise. Authors are highly appreciated your invaluable comment.
Reviewer 2, concern 2: Kindly proofread the manuscript for some typo errors.
For example: Word Correction in line 75====efficiency
And so on…
Author response: Thank you very much for your concern.
Author action: we updated the manuscript by revising and editing grammatical mistakes as shown in the revised version of the manuscript.
Reviewer2, concern 3: Few of the figures are blurry, if authors can improve their quality. For example: Fig. 8.
Author response: Thank you very much for your concern.
Author action: We updated the manuscript by redrawing the blurry figures to be in high resolution as shown in the revised version of the manuscript.
Reviewer 3 Report
1. The paper is well organized with enough details and experimental results are clearly stated. It is also to the scope of the journal.
2. Some minor issues: the small in Eq. (1) is not explained; Figure 9-11 are nowhere cited in the text.
3. The useful link length of LoRA is tested by the authors to be 590m. Is it an optimistic or conservative number when deploying the devices?
Author Response
To Reviewer 3
Comments and Suggestions for Authors
The paper is well organized with enough details and experimental results are clearly stated. It is also to the scope of the journal.
Author response: Thank you very much for summarising our paper contributions.
Author action: We updated the manuscript by addressing your invaluable comments and suggestion. Authors are highly appreciated your concern and suggestions.
Reviewer3, concern 1: Some minor issues: the small in Eq. (1) is not explained; Figure 9-11 are nowhere cited in the text.
Author response: Thank you very much for your concern.
Author action: We updated the manuscript by correction in Eq.(1); and now, Figure 9-11 are cited into the revised version of the manuscript.
Reviewer3, concern 2: The useful link length of LoRA is tested by the authors to be 590m. Is it an optimistic or conservative number when deploying the devices?
Author response: Thank you very much for your concern.
Author action: The link length of LoRa is tested by the authors and found signal to be 590m, after this range signal has been distorted.
Round 2
Reviewer 1 Report
The paper can be accepted
Reviewer 2 Report
Thank you for the revised version of the manuscript. The quality of the manuscript has been improved and can be considered for the possible further process.